# Antibacterial Cu or Zn-MOFs Based on the 1,3,5-Tris-(styryl)benzene Tricarboxylate

**DOI:** 10.3390/nano13162294

**Published:** 2023-08-09

**Authors:** Sorraya Najma Kinza Lelouche, Laura Albentosa-González, Pilar Clemente-Casares, Catalina Biglione, Antonio Rodríguez-Diéguez, Juan Tolosa Barrilero, Joaquín Calixto García-Martínez, Patricia Horcajada

**Affiliations:** 1Advanced Porous Materials Unit, IMDEA Energy Institute, Av. Ramón de la Sagra 3, Mostoles, 28935 Madrid, Spain; sorraya.lelouche@imdea.org (S.N.K.L.); catalina.biglione@imdea.org (C.B.); 2Escuela Internacional de Doctorado, Universidad Rey Juan Carlos, c/ Tulipan, s/n, Mostoles, 28933 Madrid, Spain; 3Centro Regional de Investigaciones Biomédicas (CRIB), Universidad de Castilla-La Mancha, C/Almansa 14, 02008 Albacete, Spain; laura.albentosa@uclm.es (L.A.-G.); pilar.ccasares@uclm.es (P.C.-C.); juan.tolosa@uclm.es (J.T.B.); 4Facultad de Farmacia, Universidad de Castilla-La Mancha, Av. Dr. José María Sánchez Ibáñez, s/n, 02008 Albacete, Spain; 5Departamento de Química Inorgánica, Universidad de Granada, Av. Fuentenueva s/n, 18071 Granada, Spain; antonio5@ugr.es

**Keywords:** metal–organic framework, antibacterial activity, oligo(styryl)benzenes

## Abstract

Metal–organic frameworks (MOFs) are highly versatile materials. Here, two novel MOFs, branded as IEF-23 and IEF-24 and based on an antibacterial tricarboxylate linker and zinc or copper cations, and holding antibacterial properties, are presented. The materials were synthesized by the solvothermal route and fully characterized. The antibacterial activity of IEF-23 and IEF-24 was investigated against *Staphylococcus epidermidis* and *Escherichia coli* via the agar diffusion method. These bacteria are some of the most broadly propagated pathogens and are more prone to the development of antibacterial resistance. As such, they represent an archetype to evaluate the efficiency of novel antibacterial treatments. MOFs were active against both strains, exhibiting higher activity against *Staphylococcus epidermidis*. Thus, the potential of the developed MOFs as antibacterial agents was proved.

## 1. Introduction

Bacteria are considered one of the major causes of serious infectious diseases [1]. Bacteria which were once easily treated with antibiotics have evolved and gradually become resistant to more and more treatments [2]. Thus, the fabrication and optimization of novel antibacterial compounds are urgently needed. In this regard, transition metal based materials appear to be a promising alternative because they exhibit low toxicity and a relatively low cost. They can present compelling properties as pure metal nanoparticles [3], oxide particles [4], or ions [5]. Zinc (II) and copper (II) ions are particularly interesting for antibacterial applications since they are involved in numerous biological pathways. On the one hand, zinc protects against reactive oxygen species via the antioxidative effect of cysteine-rich metallothionein, playing an important role in wound repair as a cofactor in enzymatic systems such as zinc-metalloproteinases [6]. This metal impacts glucose metabolism through the regulation of insulin expression. Furthermore, it is a considerable component of bone structure. On the other hand, copper has antioxidative properties as well and is involved in the reduction of reactive oxygen species [7]. This element is also fundamental in the synthesis of phospholipids found in myelin sheaths, which protect peripheral nerves. Lastly, it is found in lysyl oxidase, an enzyme that promotes the formation of collagen and elastin [8].

Another approach to the formulation of new antibacterial agents is employing organic dendrimers; such compounds have shown antimicrobial properties, as well as potential as bacteriophobic coatings [9]. Recent studies indicate that ionic dendrimers are more efficient against bacteria. One of the advantages of dendrimers is that the mechanisms of action can be tuned by interactions with different metal ions and light irradiation [10]. 1,3,5-tris(styryl)benzene derivatives can be visualized as a zero-generation dendrimer of poly(phenylenvinylenes). These compounds have been widely studied for their excellent optical properties that are advantageous in OLED devices [11]. Furthermore, applications in the biomedical and healthcare fields have recently been described [12,13,14].

In this context, metal–organic frameworks (MOFs) are presented as a balanced solution by combining, through coordination, active metals and organic linkers with antibacterial properties [15]. MOFs are hybrid materials composed of inorganic cations or clusters and polychelating organic linkers and constitute a family of versatile porous materials. They possess an array of chemical compositions, structures, and morphologies that are easily tunable [16]. This ensures a broad spectrum of structural and chemical properties and, in turn, numerous applications in disparate fields. As a matter of fact, they have been proposed in water decontamination [17], fluid sorption and separation [18], catalysis [19], sensing [20], fuel cells [21], and drug delivery [22], among others. Recent works have shown the potential of MOFs as antimicrobial agents [23]. For instance, an AgBDC MOF thin film was developed, showing interesting antifouling properties. This material protects the film surface from the adhesion of *E. coli* and hinders the proliferation of the bacteria [24]. On another front, two novel Zn-based MOFs were presented as Zn^+2^ reservoirs. These two materials based on 5-((4-carboxyphenyl)ethynyl) isophthalate linker have antibacterial activity against a Gram-negative and a Gram-positive strain, *Escherichia coli* and *Staphylococcus aureus*, respectively. The activity over time was attributed to the release of Zn (II) ions as a function of time [5].

Although several MOFs based on active metals have been reported, only a few examples are currently available where both the metal and linker are antimicrobial [25]. In this matter, a zinc thiabendazole MOF was tested against *Escherichia coli* and *Staphylococcus aureus*, as well as against *Aspergillus niger* and *Trichoderma virens* fungi. The material proved to be suitable for a controlled release in the medium of active compounds, both linker and metal [26]. Furthermore, some copper bipyridyl MOFs were tested against *E. coli*, *S. Aureus*, *K. pneumonia*, *P. aeruginosa*, and *Methicillin-resistant S. aureus* and demonstrated up to 99.9% bacteria death. The proposed mechanism of action is that MOFs hook onto the bacteria’s surface, after which the copper oxidizes proteins in the membrane, leading to bacteria death [27].

With this dynamic, this work presents two novel MOFs (IEF 23 and IEF-23) based on antibacterial agents: the 1,3,5-tris(styryl)benzene tricarboxylic acid (or 4,4′,4″-(benzene-1,3,5-triyltris(ethene-2,1-diyl))tribenzoic acid or tricarboxylate polyphenylvinylene; here referred to as PPV) with proven antibacterial activity [28], and antibacterial ions copper (II) and zinc (II). The materials were obtained via a solvothermal route and were fully physico-chemically characterized. Finally, their antibacterial properties were tested against Gram-positive and Gram-negative strains *Staphylococcus epidermidis* and *Escherichia coli* using the agar diffusion method, proving their activity.

## 2. Materials and Methods

Chemicals were obtained from Sigma Aldrich (CuCl_2_ 97%, Zn(NO_3_)_2_·6H_2_O 98%, TCI (*N*,*N*′-diethylformamide, DEF, with purity + 99%), Serviquimia (*N*,*N*′-dimethylformamide, DMF, 99%, Methanol HPLC grade), or VWR (ethanol absolute) without any further purification.

### 2.1. Synthesis

Synthesis of 1,3,5-tris(styryl)benzene tricarboxylic acid or 4,4′,4″-(Benzene-1,3,5-triyltris(ethene-2,1-diyl))tribenzoic acid: The protocol followed for the synthesis of the organic linker (PPV) was the same as previously described [28].

Synthesis of IEF-23: The synthesis of IEF-23 was performed by mixing 6.0 mg of CuCl_2_ (0.045 mmol) and 7.7 mg of PPV (0.015 mmol) in mixture of 0.5 mL of *N*,*N*′-diethylformamide (DEF) and 1 mL of ethanol in a 10-mL Duran^®^ glass tube. The mixture was homogenized using an ultrasound bath for 1 min. Then, it was progressively heated in an oven to 100 °C for 6 h and kept at that temperature for 48 h. The product was cooled down to room temperature (RT), and large emerald-colored cubic crystals were obtained via centrifugation (5000 rpm, 3 min). The solid was washed once with 5 mL of ethanol, centrifugated at 5000 rpm for 3 min, and kept wet. The synthesis was scaled up 10-fold using a 23 mL Teflon lined autoclave and following the same operation procedure. The proposed chemical formula for IEF-23 is [Cu_2_PPV]·0.2DEF·0.5EtOH, from which the reaction yield was calculated based on the metal (65%). Theoretical content in weight %: Cu 13.8; C 53.5; O 35.9; N 1.2; H 4.0 was correlated with the experimental ones, observed by ICP-OES in weight %: Cu 12.2%, and elemental analysis: C, 59.5; N, 1.1; H 4.3.

Synthesis of IEF-24: This solid was prepared by mixing 13.4 mg of Zn(NO_3_)_2_·6H_2_O (0.045 mmol) and 7.7 mg of PPV (0.015 mmol) in 3 mL of *N*,*N*′-dimethylformamide (DMF) in a 10 mL Duran^®^ glass tube. The mixture was heated in an oven to 100 °C for 6 h and kept at the temperature for 66 h. The product was cooled down to RT, and hexagonal yellow-colored crystals were collected via centrifugation (5000 rpm, 3 min). The solid was washed with 5 mL of ethanol, centrifugated at 5000 rpm for 3 min, and kept wet. The synthesis was scaled up 10-fold using a 23 mL Teflon-lined autoclave and following the same operation procedure. The proposed chemical formula for IEF-24 is [Zn_9_PPV_4_]·0.4DMF, from which an important yield (75%) was calculated based on the metal. Theoretical content in weight %: Zn 16.1; C 53.5; O 11.5; N 2.2; H 3.6. was correlated with the experimental ones, observed by ICP-OES in weight %: Zn 14.8; C 54.4; N 3.0; H 4.2.

### 2.2. Physicochemical Characterization

Elemental analysis experiments were performed using Thermo Scientific’s organic elemental analysis model 2000 after drying the samples for 2 h at 100 °C (Thermo Fisher Scientific Inc., Waltham, MA, USA). For Inductively Coupled Plasma Optical Emission Spectroscopy (ICP-OES), OPTIMA 7300-DV was used. Thermo gravimetric analyses (TGA) were carried out on SDT Q600 from TA instruments. Fourier transform infrared (FT-IR) spectra were collected using Thermo Fisher Scientific FTIR spectrometer Nicolet 6700 in attenuated total reflectance (ATR). Nitrogen isotherms were obtained at 77 K using micrometrics model microActive Tristar II. The samples were outgassed at 110 °C for 12 h prior to the measurement. UV-VIS (ultraviolet spectra) were collected with UV/Vis/NIR Lambda 1050 from Perkin Elmer. The surface charge was measured using nanosizer Zetasizer, Nano series Nano-ZS from Malvern Instruments. The solids were dispersed in the liquid media milliQ water at a concentration of 0.1 mg·mL^−1^ using an ultrasound tip (UP400S, Hilscher, Teltow, Germany) at 20% amplitude for 20 s. The materials were observed using an Olympus optical microscope model SZ61 and a tabletop scanning electron microscope (SEM) from Hitachi model TM-1000. Powder X-ray diffraction (PXRD) patterns were collected with a Panalytical diffractometer empyrean DY662, in Bragg Brentano geometry from 3 to 35 (2θ). The same conditions were used for the Le Bail fitting. Variable-temperature powder X-ray diffraction (VTPXRD) patterns were acquired every 10 °C with BRUKER D8 Advance A25 equipped with a LYNXEYE XE detector, operating at 40 kV and 40 mA and an Anton Paar XRK 900 high-temperature chamber (Cu K*a* X-radiation, λ = 1.5418 Å), from 3 to 30° 2θ (0.03 step size and 1 s step time) in a chamber/detector under continuous airflow of 10 mL·min^−1^ and a heat rate of 5 °C·min^−1^.

### 2.3. Antibacterial Activity

The Gram-positive *Staphylococcus epidermidis* (ATCC 14990) and Gram-negative *Escherichia coli* (ATCC 11775) bacteria were used as reference strains for the antibacterial testing. These strains were kept at −80 °C in glycerol (20% *v*/*v*) and reactivated in Luria–Bertani agar at 37 °C overnight. Liquid cultures were prepared in Luria–Bertani broth and incubated at 37 °C under stirring (200 rpm). Inoculums were diluted to 106 UFC·mL^−1^ in fresh LB broth by measuring OD at 600 nm.

The bacterial inhibition effect of 2 mg of each MOF, IEF-24, and IEF-23, along with the corresponding percentage of each precursor (PPV (49.5%) and metallic salt Zn(NO_3_)_2_.6H_2_O and CuCl_2_), were evaluated using the solid agar diffusion method in Mueller–Hinton plates. These plates were previously inoculated with 6 × 10^5^ UFC of each strain. Tested materials were then placed on the surface of the agar and incubated at 37 °C for 7 days. After this incubation, plates were digitally photographed at 1, 2, 3, 4, and 7 days, and the diameter of the zone of inhibition was measured with the KLONK Image Measurement 21.7.1. Antibacterial activity of the tested compounds was compared with that of the precursors in proportional concentrations. All experiments were repeated at least three times. Bacterial strains and procedures were tested with commercial disks impregnated with tetracycline only for control purposes. No comparison of activity was performed as the nature of the compounds is different.

## 3. Results

### 3.1. Synthesis and Physicochemical Characterization

Two novel MOFs based on the tricarboxylate PPV linker were obtained by solvothermal synthesis, one based on copper and the other on zinc, named IEF-23 and IEF-24, respectively (IEF stands for IMDEA Energy Framework; Figure 1). Summarily, IEF-23 was synthesized in a mixture of absolute ethanol and DEF 2:1, and a ratio of CuCl_2_ to PPV 3:1; the mixture was heated to 100 °C for 6 h and kept at the mentioned temperature for 48 h. Cubic emerald green crystals were obtained of up to ~800 μm, as shown in Appendix A. On the other hand, IEF-24 was prepared in pure DMF in a ratio of Zn(NO_3_)_2_·6H_2_O to PPV 1:3, heating to 100 °C and maintaining that temperature for 66 h. Large hexagonal yellow-colored crystals were obtained up to ~450 μm (Appendix A). FTIR spectra reveal the coordination of the carboxylate groups of the PPV ligand with the metals (Appendix A). Indeed, the asymmetric and symmetric vibrations of the carboxylic groups of the free linker were detected at 1687 and 1421 cm^−1^, respectively. These bands are shifted to 1647 and 1398 cm^−1^ in IEF-23 and to 1660 and 1396 cm^−1^ in IEF-24, confirming coordination with the metal. The shoulder in the 1647 and 1660 cm^−1^ band indicates the presence of coordinated DEF and DMF, respectively, thus suggesting that the metal ions present coordination with the solvent. Considering the vibration shift of the carboxylate groups, as well as the proposed formula, the coordination sphere of the metal centers can be hypothesized. In fact, numerous MOFs based on bidendate carboxylate linkers have been reported. Focusing on copper phases, the collected data for IEF-23 suggest that the copper may form dimers connected by the carboxylic groups, where each copper presents a square-based pyramid coordination [29,30,31,32,33]. The presence of DEF in the structure also seems to indicate that the coordination of the copper is potentially similar to the paddle wheel, which is a recurring building block in copper MOF structures [34]. In a similar manner, the coordination of zinc in IEF-24 can be suggested: the zinc ions may form clusters connected by the linker, thus bridging two metals [35,36,37,38]. Considering the flexibility of the material, it is possible that the clusters are connected through corner-sharing polyhedra.

Although both single crystals were analyzed by SCXRD using different measurement conditions, the crystals always lost crystallinity under the irradiation of the X-ray beam. Their instability under the X-ray beam hindered the unveiling of their crystalline structure. Nevertheless, unit cells were obtained and fitted to the PXRD patterns.

IEF-23 seems to crystallize in the cubic *Pm3m* space group (n. 221) with unit cell parameters (a~29.2 Å, V~25,000 Å^3^; Appendix A). The Le Bail fitting of the PXRD diffraction pattern proves the purity of the phase (Appendix A). The proposed chemical formula for IEF-23 is [Cu_2_PPV]·0.2DEF 0.5EtOH, from which the reaction yield was calculated based on the metal (65%). Theoretical content in weight %: Cu 13.8; C 53.5; O 35.9; N 1.2; H 4.0 was correlated with the experimental ones, observed by ICP-OES: Cu 12.2%, and elemental analysis: C, 59.5%; N, 1.1%; H 4.3%. The difference between the theoretical and experimental values could be explained by the presence of defects in the structure, which may occur upon drying—the sample for analysis (see below; stability data). Furthermore, the diffractograms present a wide background, which could indicate the presence of amorphous impurities.

On the other hand, the zinc phase IEF-24 seems to crystallize in the monoclinic space group *C2*/*c* (n. 15; a = 10.24, b = 24.59, c = 27.72 (Å), V = 6890 Å^3^; Appendix A), confirming the purity of the bulk material by Le Bail fitting of the PXRD (Appendix A). The proposed chemical formula for IEF-24 is [Zn_9_PPV_4_]·0.4DMF, from which an important yield (75%) was calculated based on the metal. Similarly, the experimental results obtained by ICP-OES and elemental analysis (%): Zn 14.8; C 54.4; N 3.0; H 4.2 match with the theoretical ones (%): Zn 16.1; C 53.5; O 11.5; N 2.2; H 3.6.

To further confirm the chemical composition, TGA of both MOFs was performed. IEF-23 (Appendix A) presents a first weight loss (4 wt.%) until 110 °C, tentatively attributed to ethanol, and a second step until 260 °C (22 wt.%), assigned to DEF. The linker’s oxidation and departure occur between 260 and 420 °C (57 wt.%). This trend is observed in materials that present high porosity [39]. The TGA residue, 18 wt.%, was identified as tenorite (Appendix A). The TGA estimated formula [Cu_2_PPV]·3DEF is in accordance with the chemical formula [Cu_2_PPV]·DEF_1.4_, proposed from ICP-OES and elemental analysis. On the other hand, IEF-24 (Appendix A) presents a progressive weight loss of 15 wt.% from RT until 250 °C. Between RT and ~250 °C, the weight loss is 15 wt.%, which can be explained by the elimination of the chemi- and/or physiosorbed DMF. After which, a steep weight loss of 62 wt.% can be observed up to 540 °C, which has been assigned to the oxidation of the organic linker. The residue of 22 wt.% was identified as zinc oxide (Appendix A). The TGA results allow us to estimate a formula [Zn_9_PPV_4.2_]·14DMF, which is in agreement with the previously proposed chemical formula [Zn_9_PPV_4_]·0.4DMF.

In both materials, we could speculate the presence of an important void space, which would explain the low stability of the materials upon the solvent removal, hindering their crystalline structure resolution by PXRD or SCXRD. Indeed, MOFs synthesized with such lengthy linkers tend to have tremendous void volume but low stability when in dry conditions [40,41,42]. Thus, N_2_ sorption experiments were performed at 77 K (Appendix A). The materials, activated overnight at 100 °C, showed Brunauer, Emmett, and Teller surface area (S_BET_) values of 170 and 10 m^2^·g^−1^ for IEF-23 and IEF-24, respectively. The resulting low porosity could be the consequence of the crystallinity loss upon desolvation.

In order to evaluate their structural stability, PXRD patterns were collected for both materials in three different stages: as-synthesized wet samples, dried samples (air-dried overnight), and dried samples resuspended overnight in the synthesis solution (Figure 2). Although the crystals retain their respective morphology upon drying, cubic for copper and hexagonal for zinc (Appendix A), the crystalline structure was lost. Nevertheless, after resuspending the dried materials in the synthesis solvent overnight at RT, the materials significantly recovered crystallinity and characteristic diffraction peaks of the relevant phase (Figure 2). Albeit characteristic peaks are observed, the intensity is rather low, and some peak broadening can be observed, indicating a partial amorphization (defects). The recovery of crystallinity is more notable in the case of IEF-23 than IEF-24. The potential high porosity and lengthy linker could explain this reversible amorphization/crystallization phenomenon upon desorption/adsorption of the solvent, as previously observed in highly porous materials [43].

Another key factor to consider is the thermal stability of the MOFs. Thus, VTPXRD was performed. In the case of the IEF-23 material (Appendix A), the crystalline structure is kept up to 120 °C when the solvent departure occurs, in agreement with the TGA results. The broadening of the first peaks as the temperature is increased is coherent with this assumption as it indicates a progressive loss of crystallinity since the evaporation of the solvent occurs between RT and 120 °C. This is when, according to the TGA, the solvents are gradually evaporated, potentially meaning that the crystallinity of the material is highly dependent on the porosity being occupied. Meanwhile, IEF-24 (Appendix A) started to degrade as soon as the temperature was raised, which could be explained by the collapse of the structure upon the solvent’s removal, suggesting the presence of a potential porosity. In this case, the loss of crystallinity happens fast, hinting that the airflow could be an accelerating factor, as the loss of crystallinity occurs before complete evaporation of the solvent until 350 °C, according to the TGA. Between 50 and 450 °C, an intermediate phase can be observed. It presents some characteristic peaks of IEF-24, as well as some new peaks at higher angles. This slow change in crystallinity is gradual and could be explained by a reorganization of the inorganic portion of the material happening during the solvent’s evaporation and oxidation of the linker from 50 to 450 °C. Moreover, the structure seems to be completely lost at 450 °C.

Considering the photoactive properties of the linker, IEF-23 and IEF-24 were additionally characterized by collecting UV-vis spectra in the reflectance mode and further compared with the free ligand. IEF-24 follows a similar trend when compared to the free linker, with a slight shift to a lower wavelength of 380 nm versus the free linker (Appendix A). The copper phase IEF-24 exhibits a main reflectance at 530 nm. The band was evaluated using Kubelka–Munk (Appendix A), finding that the free linker presents a band gap of 2.99 eV, 475 nm, whereas the zinc and copper present band gaps of 3.28 eV, 525 nm and 3.35 eV, 536 nm, respectively. The free linker absorbs blue light and the MOFs’ green light. This is in agreement with previous transmittance spectroscopic studies of the linker, with strong absorption in the ultraviolet region (326 nm) and fluorescence in the blue at 396 nm and with a quantum yield of 40% [44].

Moreover, the chemical stability of the materials was tested in different solvents. For this purpose, the materials were kept under stirring at 300 rpm at RT overnight at 10 mg.mL^−1^ (Appendix A). Despite these rough conditions, IEF-23 and IEF-24 are crystalline in relevant solvents usually employed in organic synthesis, such as amides (DMF, DEF), and they are partially crystalline in alcohols. In particular, IEF-24 is also stable to a limited extent in dichloromethane, toluene, and hexane, to a lesser extent. Both materials are relatively stable in protic solvents under the tested conditions. Only the zinc phase, IEF-24, presents an indication of stability in aprotic and apolar solvents. Also, the zeta potential of the materials was measured in water. Both MOF surfaces are negatively charged: −27.4 ± 5.6 mV for IEF-23 and −39.9 ± 6.4 mV for IEF-24. These results are promising, considering that charged materials tend to have a higher bactericide effect as they destabilize the membrane of bacteria through electrostatic interactions [45].

### 3.2. Antibacterial Activity

Considering that both constitutive units (ligand and metal) of the novel materials have already proven an antibacterial effect, their application as an antimicrobial is envisioned [28,37]. Thus, preliminary studies were performed using two of the most broadly propagated bacteria, both Gram-negative and Gram-positive. In this sense, the biocidal properties against *E. coli* and *S. epidermidis* were investigated for IEF-23 and IEF-24 using an agar diffusion test in solid form. This involved determining the inhibition halo resulting from the presence of MOFs, along with the equivalent amount of the tested concentration of metal precursor and ligand as controls. Representative images of the halo, along with their quantification, are presented in Figure 3 and Figure 4. Both MOFs exhibited potential antibiotic activity against *E. coli* and *S. epidermidis* infections, with a more pronounced effect observed on *S. epidermidis*. The ligand did not show a therapeutic effect on Gram-negative bacteria (0 mm) but was effective against Gram-positive ones (13.4 mm), which is consistent with previously observed results [28]. In the case of the metal precursors, the salts themselves exhibited activity against both types of bacteria, with a higher effect observed against Gram-negative bacteria. More specifically, Zn(NO_3_)_2_ salt shows halos of 10.0 mm for *E. coli* and *S. epidermidis*, while CuCl_2_ salt presents values of 7.7 and 5.4 mm for *E. coli* and *S. epidermidis*, respectively.

The activity of both MOFs against *E. coli* appears to be lower or similar to that of the metal precursor (IEF-24 = 9.0 mm and IEF-23 = 5.4 mm) (Figure 4). This, together with the absence of the effect of the PPV ligand against *E. coli*, suggests that the MOFs’ effect on *E. coli* is exclusively due to the metal activity. In addition, the lower activity of the MOFs when compared with the free cations might be explained by the progressive release of the cations to the media, acting as a cation reservoir.

Regarding IEF-23 and IEF-24′s efficiency for Gram-positive strains, the MOFs exhibit significantly greater activity against *S. epidermidis*. More specifically, IEF-23 shows halos of 18.6 mm and IEF-24 of 21.9 mm. Both materials showed higher inhibitions compared to their corresponding organic or inorganic precursors. However, the activity can be considered neither synergetic nor additive since they are lower than the sum of the activities of the precursors. This fact could be explained again by the progressive degradation of the materials.

Consequently, both materials demonstrated biocidal characteristics against both bacterial strains, with a more pronounced effect observed on Gram-positive bacteria. On Gram-negative bacteria, the biocidal effect is solely due to the metal, while on Gram-positive bacteria, both the metal and the linker play a role in bacterial inhibition, showing higher activity than the linker or metal alone, thus emphasizing the contribution of both the inorganic and organic component of the MOFs. In addition, for both Gram-positive and Gram-negative bacteria, the inhibition halo formed by the MOF powder remained stable for at least the duration of the 7-day experiment without retracting or becoming cloudy. This indicates that the bacteria would not proliferate in the inhibited zone once the antibiotic action ceased (Appendix A). Finally, the inhibition observed in the case of IEF-24 appeared to be slightly higher than that of IEF-23. Copper and zinc have both been reported to have the same efficacy at the same dose [46]. The difference in activity between the two materials could be explained by the lower metal content in IEF-23 with regard to IEF-24.

## 4. Conclusions

Two novel MOFs were built up of antibacterial active building blocks, both organic (PPV) and inorganic (Cu or Zn). Both [Cu_2_PPV]· *n*solv. (IEF-23) and [Zn_9_PPV_4_]·*n*solv. (IEF-24) display reversible loss of crystallinity and porosity upon solvation/desolvation. In addition, the materials demonstrated antimicrobial activity against both Gram-positive (*S. epidermidis*) and Gram-negative (*E. coli*) bacteria. In the case of *E. coli*, the activity is solely an effect of the metal in the material structure, whereas against *S. epidermidis*, the activity arises from the organic linker as well as the metals in the structure, thus showing greater activity than the sole precursors.

## Figures and Tables

**Figure 1 nanomaterials-13-02294-f001:**
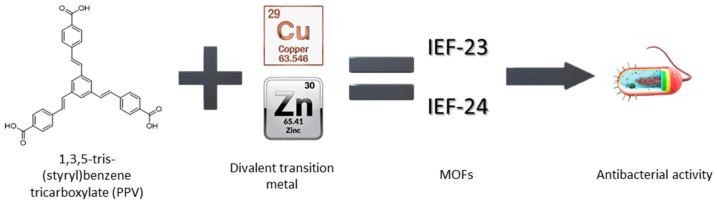
Summary scheme of results.

**Figure 2 nanomaterials-13-02294-f002:**
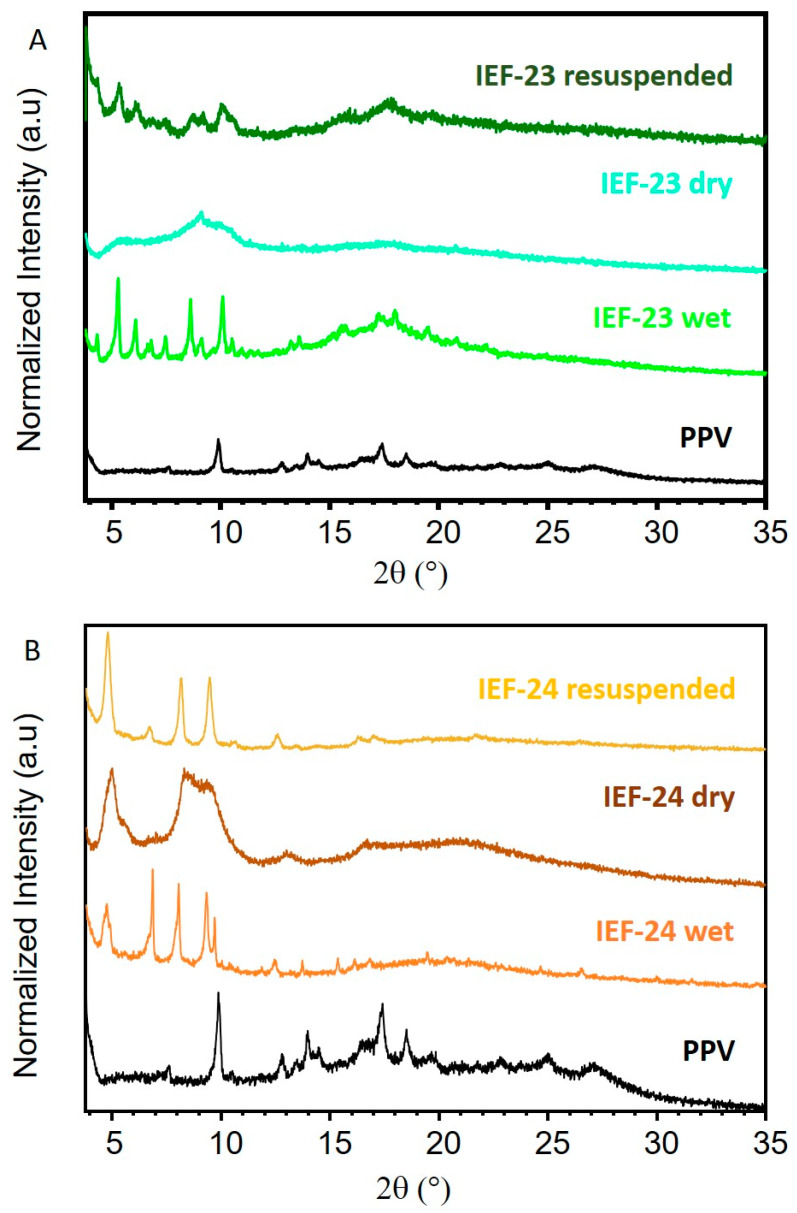
PXRD patterns of (**A**) PPV, IEF-23 wet, dry, and resuspended in DEF:EtOH 1:2, and (**B**) PPV, IEF-24 wet, dry, and resuspended in DMF.

**Figure 3 nanomaterials-13-02294-f003:**
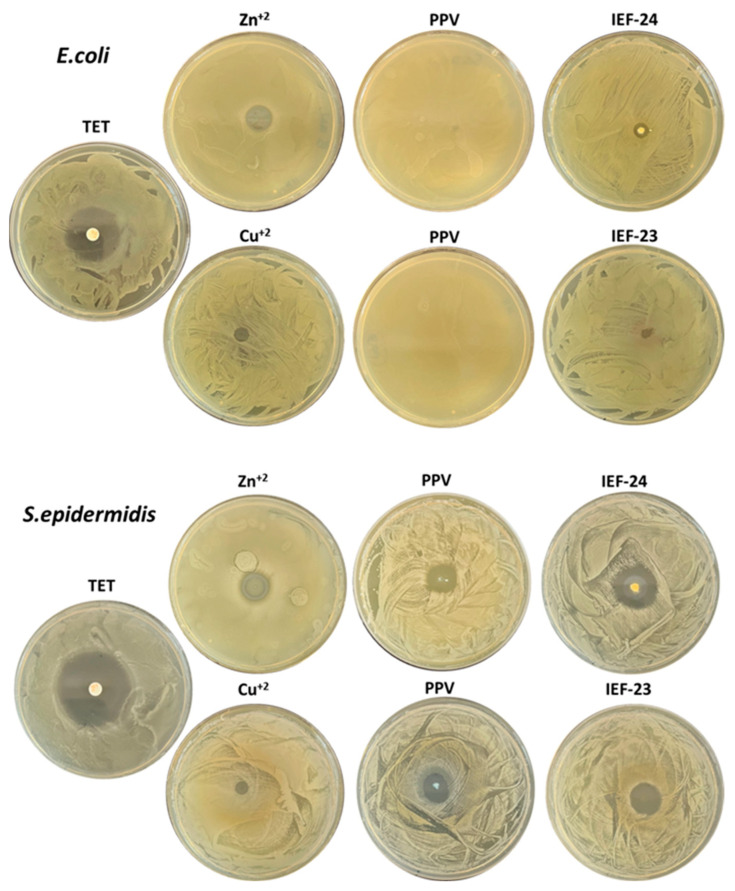
Antibacterial activity tested using solid agar diffusion method. Representative results for *E. coli* and *S. epidermidis* after 24 h of incubation at 37 °C are shown.

**Figure 4 nanomaterials-13-02294-f004:**
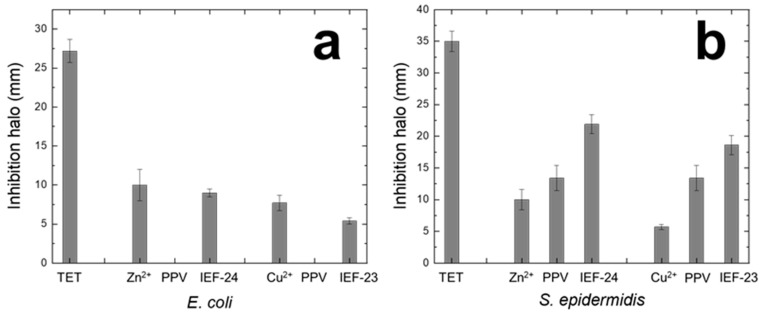
Inhibition halo (mm; determined by KLONK) against (**a**) *E. coli* and (**b**) *S. epidermidis* after 24 h exposure.

## Data Availability

Datasets will be available in Zenodo, publicly accessible repository.

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
