# Peer review of "Antibacterial Cu or Zn-MOFs Based on the 1,3,5-Tris-(styryl)benzene Tricarboxylate"

_nanomaterials, 2023, doi:10.3390/nano13162294_

Round 1

Reviewer 1 Report

The authors reported two novel MOFs and tested their antibacterial performance, which proved the possibility that MOFs can serve as potential antibacterial agents. This report is very interesting and timely, which promotes the potential, real application of MOFs in real scenarios. The manuscript is well organized with all data presented in order. This reviewer suggests the acceptance of this manuscript as it is.

Reviewer 3 Report

In this manuscript entitled "Antibacterial Cu or Zn-MOFs based on the 1,3,5-tris-(styryl)benzene tricarboxylate", the authors describe the synthesis, physicochemical properties, and antibacterial activity of two novel MOFs (IEF-23 and IEF-24).

This article is interesting, and the study is relevant. Bacterial infections have a large impact on public health. Bacterial resistance to antibiotics is a growing concern. I believe that the manuscript is suitable for publication in Nanomaterials after the major revision.

Comments and questions:

1. The English in the article needs correction.

2. The format of the references at the end of the article does not comply with MDPI standards. It is required to bring them into conformity with the format used by the publisher.

3. References in the text of the article should be placed before punctuation marks ([]. and [],).

4. The names of bacterial strains are not italicized in some places. Correction required.

5. My main concern is Section 3.2 regarding biological activity. For some reason, suddenly, for tetracycline (TET), the disk diffusion method is used, although for the rest of the studied compounds, the solid agar method is used. Accordingly, it is incorrect to compare them (there is also no comparison in the text). However, of course, a standard for such studies is needed. In addition, there are simply photos of Petri dishes. Zones of inhibition are not given in the article, although it would be more logical for them to be present there than in the ESI. More research or explanations and analytics are needed here.

Moderate editing of English language required. Please check all the text carefully and make corrections. For example:

line 20: ...by agar the diffusion method.

line 24: ...as antibacterial agents was proven.

line 28: Bacteria are considered...

line 29: Bacteria that were once...

line 31: ...are urgently needed...

lines 32-33: ...and a relatively low cost.

and so on.

Round 2

Reviewer 2 Report

line 35: not zinc and copper, but zinc(II) and copper(II) ions are

line 183: not metal, but the metal ion presents

average

Author Response

We would like to thank the reviewer for the suggestions, those corrections have been introduced in the final manuscript. Also, we acknowledge the reviewer for  taking time to review our manuscript.

Reviewer 3 Report

The authors have made a large number of changes to the article. The content of the article has improved significantly. I propose to accept it for publication in its current form.

Author Response

We would like to thank the reviewer for the feedback and constructive suggestions, and for taking the time to review our manuscript.